# Prevalence and genome features of lake sinai virus isolated from *Apis mellifera* in the Republic of Korea

**Thi-Thu Nguyen[1,2☯], Mi-Sun Yoo[1☯], A-Tai Truong[1,3], So Youn Youn[1], Dong-Ho Kim[1], Se-Ji Lee[1], Soon-Seek Yoon[1], Yun Sang Cho[1] ***

1 Department of Animal and Plant Health Research, Laboratory of Parasitic and Honeybee Diseases, Bacterial Disease Division, Animal and Plant Quarantine Agency, Gimcheon, Republic of Korea, 2 Institute of Biotechnology, Vietnam Academy of Science & Technology, Ha Noi, Viet Nam, 3 Faculty of Biotechnology, Thai Nguyen University of Sciences, Thai Nguyen, Viet Nam

☯ These authors contributed equally to this work.
* choys@korea.kr

**Data Availability Statement:** All relevant data are within the manuscript and its Supporting Information files.

## Abstract

Lake Sinai Virus (LSV) is an emerging pathogen known to affect the honeybee (*Apis mellifera*). However, its prevalence and genomic characteristics in the Republic of Korea (ROK) remain unexplored. This study aimed to assess the prevalence of and analyze the LSVs by examining 266 honeybee samples from the ROK. Our findings revealed that LSV exhibited the highest infection rate among the pathogens observed in Korean apiaries, particularly during the reported period of severe winter loss (SWL) in *A. mellifera* apiaries in 2022. Three LSV genotypes– 2, 3, and 4 –were identified using RNA-dependent RNA polymerase gene analysis. Importantly, the infection rates of LSV2 (65.2%) and LSV3 (73.3%) were significantly higher in colonies experiencing SWL than in those experiencing normal winter loss (NWL) (*p* < 0.03). Furthermore, this study provides the first near-complete genome sequences of the Korean LSV2, LSV3, and LSV4 strains, comprising 5,759, 6,040, and 5,985 nt, respectively. Phylogenetic analysis based on these near-complete genome sequences demonstrated a close relationship between LSVs in the ROK and China. The high LSV infection rate in colonies experiencing a heightened mortality rate during winter suggests that this pathogen might contribute to SWL in ROK. Moreover, the genomic characteristic information on LSVs in this study holds immense potential for epidemiological information and the selection of specific genes suitable for preventing and treating LSV, including the promising utilization of RNA interference medicine in the future.

## Introduction

Honeybee (*Apis mellifera*) pathogens play a crucial role in colony decline, with some viruses causing symptomatic infections, whereas others remain uncharacterized [1–3]. Among these, the Lake Sinai Virus (LSV) is prevalent in weak or collapsed colonies [4]. However, the specific symptoms of LSV infection remain unclear, posing challenges to its detection in apiaries [4]. LSV infections are often observed alongside other viral and bacterial infections, leading to

**Funding:** YSC N-1543081-2021-25-03 Animal and Plant Quarantine Agency https://www.qia.go.kr/listindexWebAction.do The funder had no role in study design, data collection and analysis, decision to publish, or preparation of the manuscript.

colony loss [2]. It has been recently reported to affect honeybees [4–7], LSVs were first discovered in the United States (US) surveys [2,7] and subsequently demonstrated to be widespread in the US [4,8,9]. Since then, LSV strains have been widely reported worldwide, including in Spain, Belgium, Türkiye, Germany, Australia, Slovenia, China, Japan, and the Republic of Korea (ROK) [2,10–18]. However, limited information exists regarding LSV infections in Korean apiaries. Therefore, determining the presence of LSV in apiaries suffering from severe winter loss (SWL) is helpful for understanding the influence of this pathogen in ROK.

LSV is a positive-sense single-stranded RNA virus. The genome characteristics of LSV encompass three main genes: 1) the first non-structural protein (NS1), 2) RNA-dependent RNA polymerase (RdRp), and 3) Capsid protein (CP) [4,7,19,20]. Additionally, the genome contains a region for the second non-structural protein (NS2). The NS1 (also known as open reading frames 1 (ORF1)) has an unclear function [19]. RdRp regions (ORF2) of LSV are responsible for viral RNA replication, making it essential for viral propagation [5]. The capsid region (ORF3) plays a crucial role in the virus lifecycle, as it recognizes the host and assembles the virus capsid [19]. Meanwhile, the NS2 region (ORF4) can potentially facilitate infection in arthropod hosts, but its specific role remains under investigation [21]. Continuous molecular surveys are underway to assess the relative conservation of various LSV strains due to the significant diversity observed among them. Presently, species-specific RdRp phylogenetic analyses have led to the identification of two phylogenetic clusters known as LSV1 and LSV2 [7]. Furthermore, there exist strains labeled as sister strains, potentially representing recombinants of LSV1 and LSV2, with designations on the National Center for Biotechnology (NCBI) such as LSV3–LSV8, LSV NE, LSV SA1, LSV SA2, LSV TO. Discrimination among these strains relies heavily on molecular surveys, including PCR-based methods, RT–PCR, real–time RT–PCR, and metagenomics. It is noteworthy that the primers utilized for variant detection have, thus far, not exhibited sensitivity to all known genotypes [3,4,7,9,22]. Notably, Faurot-Daniels et al. found a correlation between LSV2 prevalence and high colony loss in honeybees [23], suggesting that this virus contributes to colony collapse [2,4,5,20,22]. Although the pathogenicity and specific symptoms of LSV infection in colonies have not been well defined [1–3], the pathogen is assumed to weaken the immune system of honeybees, making them more susceptible to other pathogens and stressors [19]. Due to the lack of symptomatic features of LSV in honeybees, molecular biological diagnosis becomes imperative for detecting the infection levels of this virus within honeybee apiaries. LSV is not only detected in honey bees but has also been identified in certain ant species [20], on pollen loads and *Varroa destructor* [4], and wild bumblebee hosts (namely *Bombus pascuorum*, *Bombus lapidaries*, *Bombus pratorum*, *Bombus atratus* [24,25], sweat bee (*Halictus ligatus)* [9], solitary bees (*Andrena vaga*, *Osmia bicornis*, *Osmia cornuta)* [26], mining bee (*Andrena spp.*) [3,27], and Vespids (Hornet) [28]. Furthermore, studies have identified LSVs in honeybee population experiencing colony collapse disorder and weakened hive conditions [2,4,6,23]. Interestingly, Daughenbaugh et al. [4] have observed discrepancies in the detection rates of LSVs between strong and weak colonies, potentially highlighting a complex association between viral presence and colony health. Therefore, understanding the prevalence of LSV in honeybee colonies is vital for identifying the factors contributing to colony collapse disorder.

Accordingly, the present study aimed to detect LSV variants (LSV2, LSV3, and LSV4) in honeybees collected from ROK and analyze the characteristic near-complete genome sequences of the identified LSV variants (LSV2, LSV3, and LSV4) within the country. This study reported the utilization of a shared LSV primer pair for variant detection and investigated the relationship between LSV variants and SWL. Furthermore, phylogenetic analyses were performed to establish the relationship between Korean LSVs and those detected in other countries.

## Materials and methods

### Collection of honeybee samples

The worker bees ($n$ = 10~30) were collected from each of 266 honeybee colonies from 137 apiaries in different provinces of ROK between January and August 2022. The honeybee samples in this study were collected based on the observed decline in the vitality of adult honeybees and an unusual decrease in the population. The symptoms within honeybee colonies corresponded to the phenomenon of SWL or Colony Collapse Disorder (CCD). The number of honeybee samples collected depended on the number of lost honeybee colonies in each apiary. These honeybee colonies in apiaries are chiefly dedicated to honey harvesting, typically remaining fixed at a specific location. Among them, 141 honeybee samples from 57 apiaries in 5 provinces were collected from January to March 2022, of which 115 and 26 were suffering from SWL and normal winter loss (NWL), respectively (S1 Table). The remaining 125 samples collected from April to August were obtained from healthy colonies. Live samples were collected in 50–mL falcon tubes, and the samples were transferred to a laboratory and stored at –80˚C for further analysis. This study did not require ethical approval since it did not include vertebrates or cephalopods.

### Total nucleic acid extraction and detection of honeybee pathogens

The total nucleic acids of the honeybee samples were extracted following a previously reported method [29] using a Maxwell® RSC viral total nucleic acid purification kit (Promega, Madison, WI, USA). First-strand complementary DNA (cDNA) was generated from nucleic acids using GoScript™ Reverse Transcriptase (Promega) in conjunction with an oligo(dT) primer, according to the manufacturer's instructions.

The presence of LSV2 in honeybee samples was confirmed by amplification of the RdRp gene from cDNA using primer pair LSV-For:5′–GCTTGTCGTGGATTCTGGTC–3′ and LSV-Rev:5′–CTCAGCACGAAATCGCTCAA–3′. The primer pairs were designed based on the LSV2 sequence (NCBI accession number: HQ888865.2). Positive detection of LSV2, LSV3, and LSV4 was confirmed using three genotype-specific primer pairs (S1 Fig and S2 Table). Finally, a sequence analysis was performed using the RdRp region of each genotype. PCR was performed using AccuPower® PCR PreMix (Bioneer, Daejeon, ROK). The 20-μL reaction mix comprised 2 μL cDNA template (20 ng/μL), 1 μL of each primer (10 pmol), and 16 μL ddH$_2$O. The PCR reaction conditions were as follows: 94˚C for 2 min, followed by 30 cycles of 94˚C for 30 s, 60˚C for 30 s, and 72˚C for 1 min, followed by a final elongation step at 72˚C for 10 min. The amplified PCR products were resolved by 1% agarose gel electrophoresis. DNA bands were stained with ethidium bromide (0.5 μg/mL) and visualized under UV light. The fragment was purified using a QIAquick® gel extraction kit (Qiagen, Hilden, Germany) according to the manufacturer's instructions. LSVs sequencing was performed using a Sanger sequencing instrument (Macrogen, Seoul, ROK) and analyzed using BioEdit 7 and ClustalX software (Informer Technologies, Inc., Los Angeles, CA, USA) [30,31]. Specific primers for detecting each LSV genotype (LSV2, LSV3, and LSV4) (S2 Table) were designed for real–time RT–PCR. The real-time RT–PCR reaction mix comprised 10 μL of 2× SYBR Mix, 0.5 μL (10 pmol) of each primer, 1 μL of template DNA (20 ng/μL), and 8 μL of ddH$_2$O. The optimal cycling conditions were as follows: an initial denaturation step at 94˚C for 2 min, followed by 40 cycles of 94˚C for 30 s, 55˚C for 30 s, and 72˚C for 30 s. A melting curve dissociation analysis was performed to verify the specificity of PCR amplification. These positive controls utilized in this study were constructed through the amplification of RpRd gene of LSVs (LSV2, LSV3, and LSV4) from Korean honeybees, subsequently cloned into pGem–T vector and

transferred to *Escherichia coli* DH5α cells. Negative and positive controls were included for each run. All experiments were performed in triplicate for each sample. Samples with a $C_t$ value $\leq 35$ and consistent melting curves were considered positive. To determine the correlation between LSVs (LSV2, LSV3, and LSV4) and SWL, we investigated the presence of LSVs in 141 honeybee colonies collected from 54 apiaries from January to March 2022, of which 115 were identified as having SWL and 26 with NWL. The LSV2 surveillance was then extended to another 125 honeybee colonies collected from April to August 2022

### Sequence analysis and LSVs genome assembly

The nucleotide sequences of LSVs (LSV2, LSV3, and LSV4) were searched and compared to nucleotide sequence databases using the NCBI nucleotide Basic Local Alignment Search Tool (BLASTn). After confirming the presence of LSVs (LSV2, LSV3, and LSV4) using the detection primer pair described above, cDNA was used to amplify different fragments of the near-complete genomes of LSV2, LSV3, and LSV4 (Fig 1 and S3 Table). The alignment was generated and trimmed to the nucleotide sequences of LSVs (LSV2, LSV3, and LSV4) in BioEdit program [32]. Sequence assembly for each genotype was performed by comparing LSV2, LSV3, and LSV4 reference sequences with the NCBI accession numbers LR655824.1, MZ821900.1, and MZ821852.1, respectively.

### Phylogenetic tree analysis

Phylogenetic analysis was performed using the overlapping RdRp region (460 nt) of LSV3 and LSV4, as previously described [18]. The overlapping RdRp region were alignment and edited

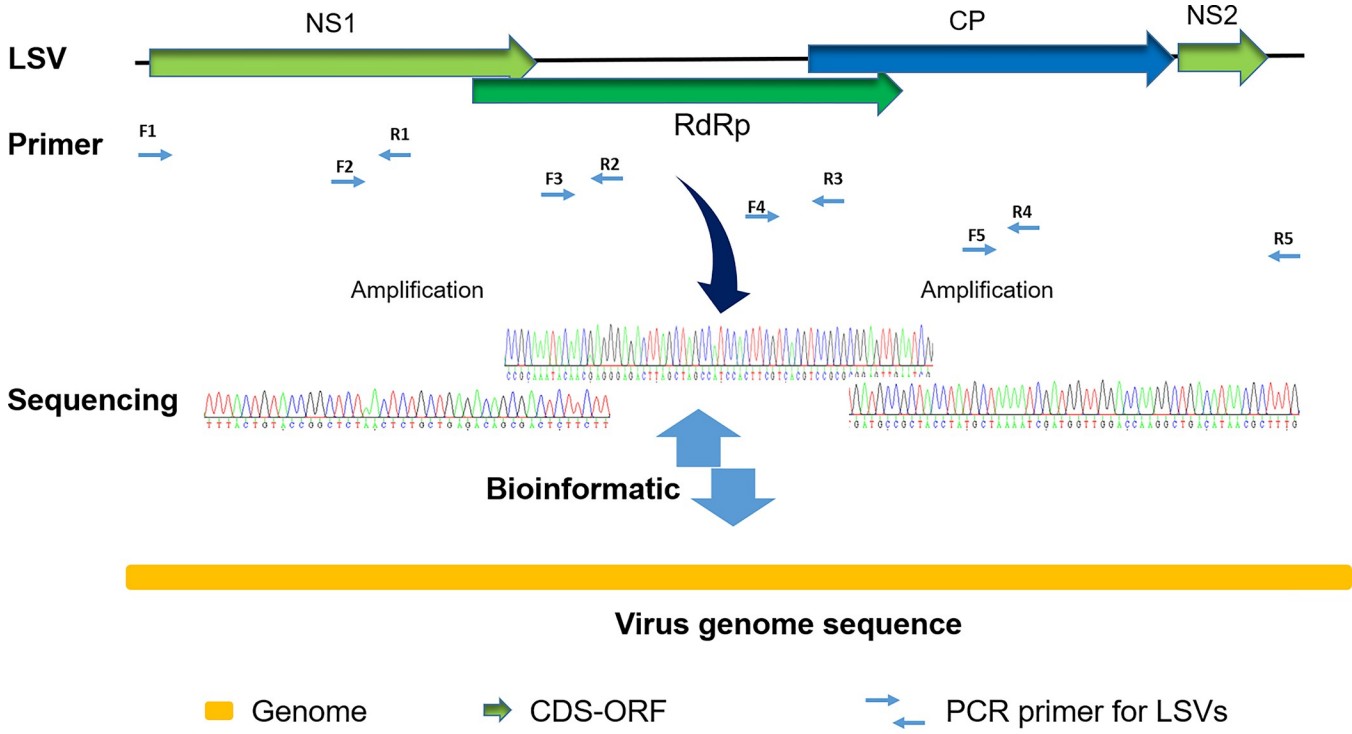

**Fig 1. Amplified fragments based on the alignment of Lake Sinai Viruses.** NS1, first nonstructural protein region; RdRp, RNA-dependent RNA polymerases; CP, Capsid protein; NS2, second nonstructural protein region. F1 to F5 denote forward primers, and R1 to R5 denote reverse primers. The details of primers are shown in S3 Table. CDS, coding sequences; ORF, open reading frame; LSVs, Lake Sinai Viruses.

and calculated after a complete alignment using the BioEdit version 7 [32]. Multiple alignments were performed using ClustalX [33]. Evolutionary distances were calculated using the Kimura two-parameter model [34]. Phylogenetic trees of LSVs were constructed based on the RdRp gene using the neighbor-joining method [35], with bootstrap values based on 1,000 replications in Mega X [36].

## Data analysis

The likelihood ratio chi-square test of contingency was applied to compute the probability of equal pathogen incidence in SWL and NWL, and the prevalence of LSV2, LSV3, and LSV4 across different sampled regions was compared using the Kruskal-Wallis H test. Statistical significance was considered at $p < 0.05$.

## Results

### Detection of LSVs in Korean apiaries

LSV2 was detected in 266 honeybee colonies was 60.9% (162/266) in ROK. A comparison of the presence of LSV2 in different provinces of ROK revealed a statistically significant variation in LSV2 infection rates ($p = 0.011$). Among the LSV2-positive samples, the highest prevalence of LSV2 was observed in Gyeongsangbuk-do (32.5%), followed by Gyeongsangnam-do (21.6%), and Jeollanam-do (17.3%) (Fig 2).

Notably, when specific primers targeting LSV2 with a target size of 218 bp were employed, additional bands of varying sizes of 460 and 1,000 bp appeared in some samples on a 1% agarose gel (S2 Fig). Sequence analysis revealed that the identified sequences exhibited the greatest resemblance to LSV2 (94.5%; MZ281853), followed by LSV3 (97.6%; MZ821878) and LSV4 (98.4%; MZ281893) (S3 Fig). This lower percent of identified sequences for LSV2 compared to the others is likely due to several nucleotide mutations within the short amplified region (218 bp) compared to the reference sequence in NCBI. The RdRp region of each genotype was amplified for phylogenetic analysis using the 460 nt RdRp region of the detected LSV in the ROK, confirming that the LSV in Korean apiaries belonged to three different genotypes: LSV2, LSV3, and LSV4 (Fig 3). LSV2, LSV3, and LSV4 in the ROK were in the same cluster as lineages originating from China and Japan rather than those observed in European countries (Fig 3). Additionally, the nucleotide identities of the RdRp-encoding gene of LSV2 were 73.2 and 73.8% relative to LSV3 and LSV4, respectively. Additionally, the similarity between the RdRp-encoding genes of LSV3 and LSV4 was 76.9%.

### Prevalence of LSVs in severe and normal winter loss

A comparison of the LSVs (LSV2, LSV3, and LSV4) in SWL and NWL was conducted in 141 honeybee samples (SWL: $n = 115$ and NWL: $n = 26$). The analysis revealed a significant difference in the infection rates of LSV2 ($p = 0.031$) and LSV3 ($p = 0.001$) between the samples from colonies with SWL and those with NWL. However, no significant difference was observed in LSV4 between the SWL and NWL samples ($p = 0.456$). The LSV genotypes among the SWL colonies were as follows: LSV2 (65.2%), LSV3 (73.9%), and LSV4 (30.4%). In contrast, NWL samples exhibited the following distributions: LSV2 (42.3%), LSV3 (38.5%), and LSV4 (23.1%) (Fig 4). The overall LSV infection rate was 83.3% (96/115) in the SWL colonies and 69.2% (18/26) in the NWL colonies.

In 2022, five of the nine provinces in the ROK had information on SWL. All three LSV genotypes were identified in four provinces: Jeollanam-do, Jeollabuk-do, Gyeongsangnam-do, and Gyeongsangbuk-do, whereas only LSV2 was detected in Jeju-do (Fig 5). Based on

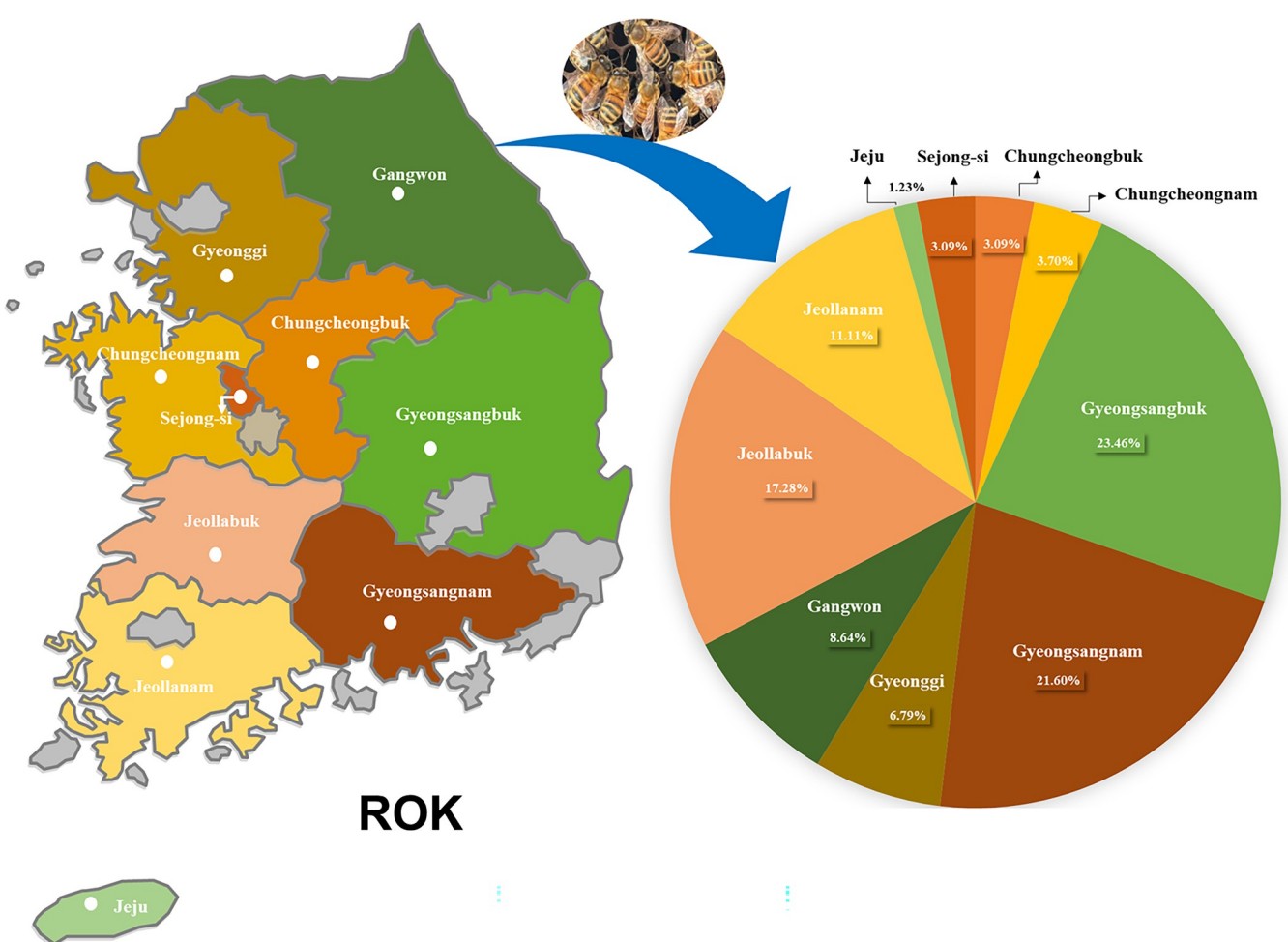

**Fig 2. Provincial distribution of Lake Sinai Virus 2 in LSV2-positive honeybee colonies in ROK.** The LSV2 isolated from honeybee samples between January and August 2022, according to sampling provinces in the Republic of Korea (ROK). Provinces are shaded according to the percentage of colonies testing positive for LSV2.

observational data, two colonies exhibited a 100% LSV2 infection rate in Jeju-do. In Jeolla-nam-do, out of 25 colonies, the infection rates for LSV2, LSV3, and LSV4 were 48.0, 44.0, and 32.0%, respectively. Among the 35 colonies from Jeollabuk-do, the corresponding infection rates of LSV2, LSV3, and LSV4 were 51.4, 77.1, and 37.1%, respectively. In Gyeongsangnam-do, out of 32 colonies, the infection rates for LSV2, LSV3, and LSV4 were 84.4, 96.9, and 15.6%, respectively. Among the 21 colonies in Gyeongsangbuk-do, LSV2, LSV3, and LSV4 infection rates were 90.5, 95.2, and 61.9%, respectively. The Kruskal-Wallis H test indicated that there is a non-significant difference between the different provinces ($p = 0.497$).

## Characteristic genome analysis of LSV

The near-complete genome of the three genotypes LSV2, LSV3, and LSV4 were sequenced and assembled; their near-complete genome sequences contained 5,759, 5,792, and 5,985 nt, respectively.

The phylogenetic variation is consistent with nucleotide similarity among the isolates. In the first main branch, the near-complete genome sequences of LSV2, LSV3 and LSV4 isolate from Korean honeybees were similarity with group isolate to China. As shown in the tree, the

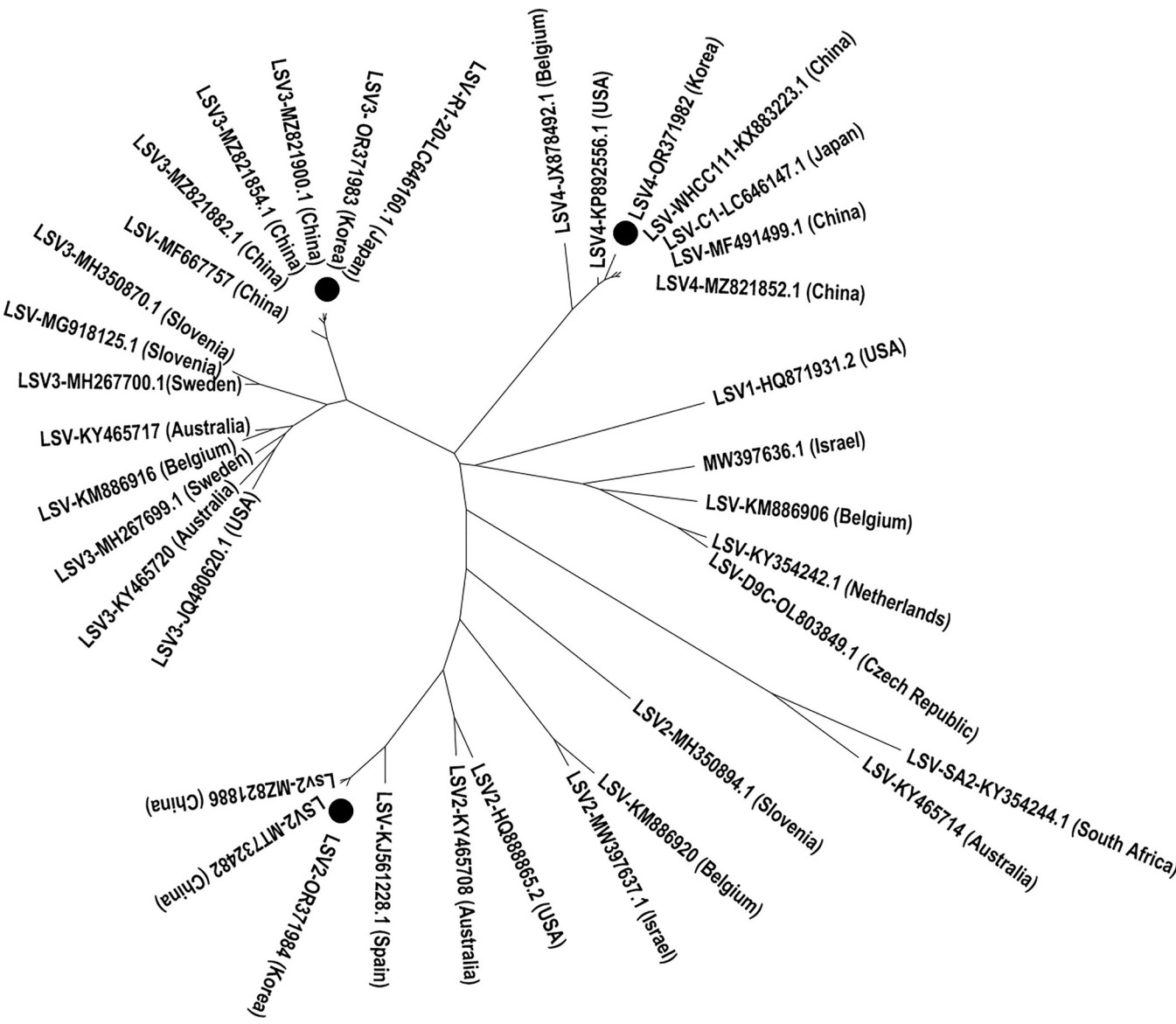

**Fig 3. Neighbor-joining phylogenetic tree of Lake Sinai Viruses based on RNA-dependent RNA polymerase gene.** The tree was constructed using 460 nt of the RNA-dependent RNA polymerase (RdRp) region of the detected Lake Sinai Viruses (LSVs) and the NCBI reference sequences of different genotypes. The strains identified in this study–LSV2, LSV3, and LSV4 –are marked with a black dot. Genotype name, NCBI accession number, and country name of each isolate are shown.

LSV isolates from ROK had a closer genetic relationship with a species of LSV isolated from China (Fig 6).

The near-complete genome feature of LSV2 comprised 5,759 nt, containing the first non-structural protein segment (NS1; from 39 to 2,576 nt), which overlapped with the RdRp segment (RdRp; from 1,813 to 3,681 nt), capsid protein segment (CP; from 3,700 to 5,262 nt), and the second nonstructural protein segment (NS2; from 5,307 to 5,759 nt) (S4 Table). LSV2 isolates had higher nucleotide similarity (96.4–98.6%) with Chinese isolates than with isolates from other countries (92.1–94.5%). The nucleotide length of NS1 in LSV2 was 2,538 nt, differing from those of other isolates, such as MZ821853.1 from China, KY465707.1 and KY465706.1 from Australia, KY354241.1 and NC035116.1 from Tonga, and OL803840.1 from

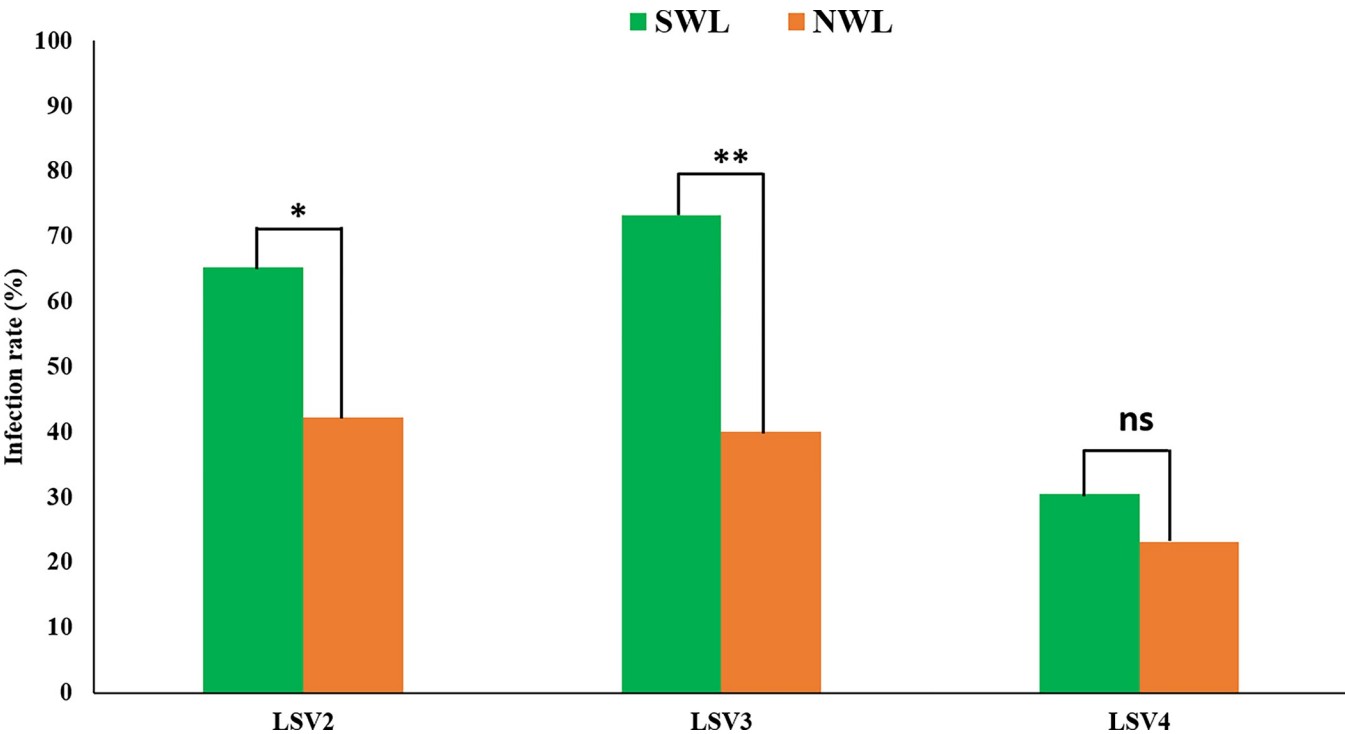

**Fig 4. Infection rate of Lake Sinai Viruses from severe and normal winter loss colonies.** Honeybee samples were obtained from 141 honeybee colonies in 57 apiaries in the Republic of Korea. Statistical comparison of the infection rate of each LSV genotype between severe winter loss (SWL) and normal winter loss (NWL) colonies was conducted using likelihood ratio chi-square test analysis. * and ** indicate significant differences at $p < 0.05$ and $p < 0.01$, respectively. "ns" indicates not significant differences.

the Czech Republic (S4 Table). The near-complete genome feature of LSV3 contained NS1 (31–2619 nt), which overlapped with RdRp (1,847–3,715 nt), CP (3,736–5,292 nt), and NS2 (5,335–5,787 nt) (S5 Table). The genome sequence of LSV3 aligned with another genome sequence of LSV3 deposited in GenBank, ranging from 84.5 to 98.7%. Multiple sequence comparisons revealed that the Korean LSV3 sequence was similar to that from China. The heterogeneity of the LSV3 genome sequence was reflected in the variance in the sequence identity of different genes: 82.2–98.7% for NS1, 85.0–98.5% for RdRp, 81.7–97.9% for CP, and 88.2–98.7% for NS2 (S5 Table). The structure of the LSV4 genome sequence was identified and included NS1 with a size of 2,549 nt (67–2,616 nt), RdRp (1,844–3,712 nt), CP (3,736–5,292 nt), and NS2 (5,334–5,786 nt). The near-complete genome sequence of LSV4 showed heterogeneity in the variance of sequence identity of different genes: 90.4–97.7% for NS1, 91.5–98.1% for RdRp, 90.3–97.9% for CP, and 90.3–98.2% for NS2 (S6 Table).

## Discussion

This study was conducted to identify the prevalence of LSVs (LSV2, LSV3, and LSV4) in Korean apiaries, analyze the near-complete genome features of three LSV genotypes in the ROK, and compare the infection rates of LSV genotypes in SWL and NWL. The infection rate of LSV2 in the ROK was remarkably high (60.9% of the collected samples). The LSV2 infection rate was considerably higher than those of other prevalent honeybee pathogens in ROK, including *Nosema ceranae*, deformed wing virus, sacbrood virus, and black queen cell virus [29]. These results suggest that LSV2 and LSV3 may significantly impact honeybee health and

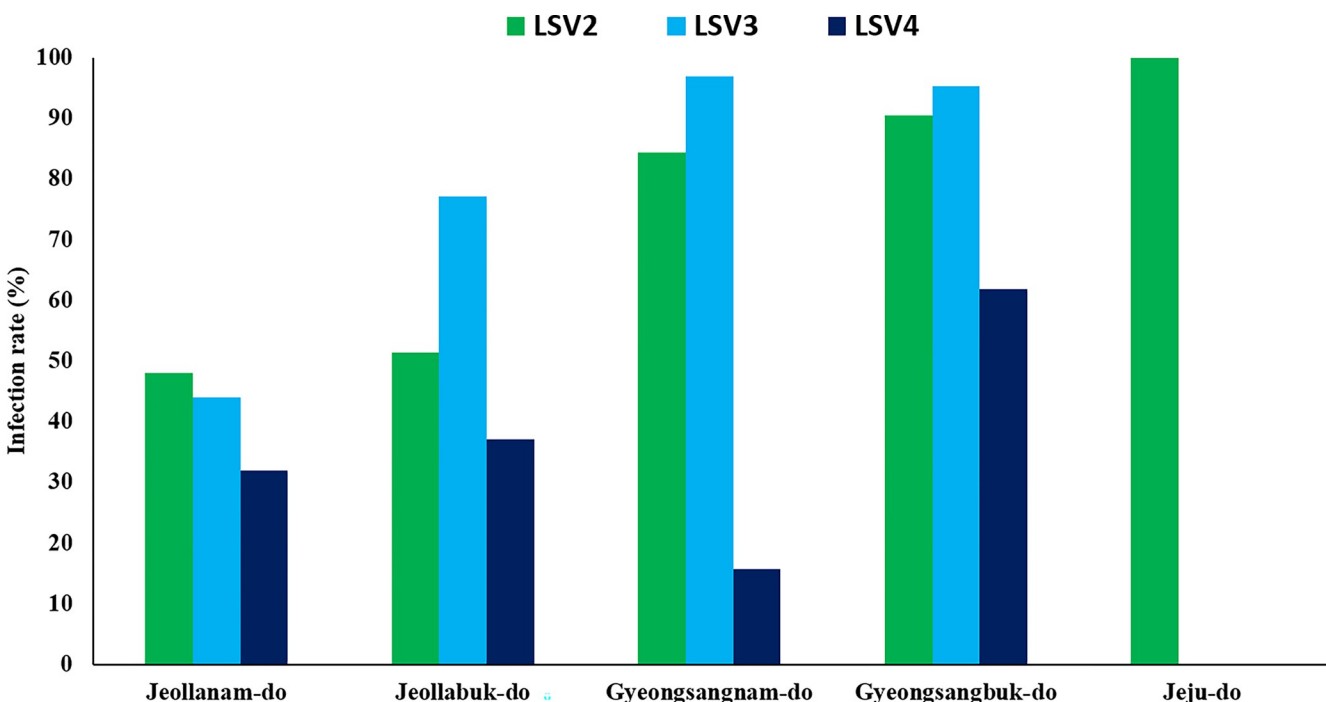

**Fig 5. Lake Sinai Virus prevalence from severe winter loss in Korean provinces.** LSV2, Lake Sinai Virus 2; LSV3, Lake Sinai Virus 3; LSV4, Lake Sinai Virus 4.

influence colony decline in ROK, and control measures for this novel pathogen in Korean apiaries are needed.

The LSVs (LSV1 and LSV2) were first detected and described in honeybee colonies in the US [7]. Subsequently, its occurrence has been reported about the prevalence of LSV variants in honeybee populations in many countries, including Germany, Australia, Slovenia, the Czech Republic, Africa, Iran, and Japan [4,11,12,15,18,35,36]. While investigating the presence of LSV1 swarms within our study, we did not detect the virus in any of the collected honeybee samples. Notable, other LSV variants including LSV2, LSV3, and LSV4 were successfully identified throughout the surveyed honeybee apiaries. This observation aligns with findings from Kwon et al. [16], which employed a metagenomic approach to uncover diverse viral elements present in Korean honeybee colonies. Hou et al. [37] reported the absence of strain C (belong to LSV1 sister group) in honeybees in China to date. This highlights the potential diversity of LSV strains, which may have implications for honeybee health and disease management. The detection of LSV3 and LSV4 using LSV2-specific primers could be attributed to PCR detection bias during intermediary sapling or primer design. In the current study, the total LSV2 infection rate in honeybee samples from the ROK was 60.9% (162/266). Honeybee colonies exhibiting signs of SWL in ROK demonstrated a remarkably high rate of positive LSVs (LSV2, LSV3, and LSV4) detection, reaching approximately 83.3% of the total detected cases. This underscores the urgent need to investigate the origin of this disease and implement appropriate measures to control LSV in Korean apiaries.

Previous research by Ravoet et al. [38] suggested that LSV might be present in pollen and *Varroa* mites. Experimental evidence has demonstrated the potential for cross-species transmission of honeybee viruses, including LSV, via interactions with *V. destructor* mites [8,39,40].

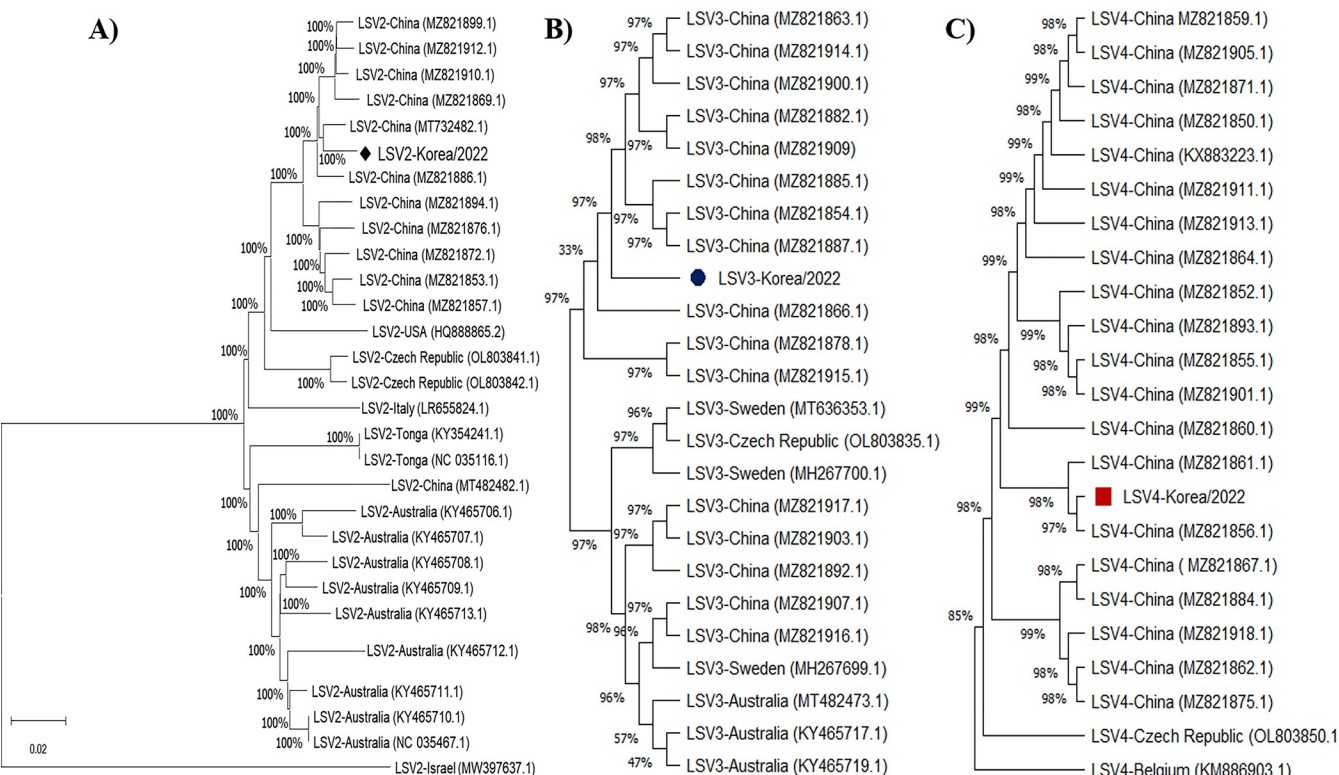

**Fig 6. Neighbor-joining phylogenetic tree of Lake Sinai Virus based on the near-complete genome feature.** The optimal tree is shown. The evolutionary distances were computed using the Kimura 2-parameter method. A, LSV2 genome feature (5,759 nt); B, LSV3 genome feature (6,040 nt); C, LSV4 genome feature (5,985 nt). (References strains in NCBI were shown in S4–S6 Tables).

These mites have been implicated as a vector in the transmission of various viruses in honeybees, such as Israeli acute paralysis virus, deformed wing virus, Kashmir bee virus, *V. destructor* virus-1, black queen cell virus, and LSV [4,15,35,41–43]. This study showed the presence of LSV2 and LSV3 in *V. destructor* mites associated with honeybee colonies but failed to detect LSV4 in any mite samples across the twelve surveyed honeybee colonies (S7 Table). This could be because LSV4 exists at very low concentrations within mites, making detection more challenging than for the identified LSV2 and LSV3 strains. Alternatively, the LSV4 might exhibit uneven distribution within bee colonies, leading to an inconsistent presence in mite populations. Additionally, the chosen sampling methodology might not have been sufficiently sensitive to detect LSV4, requiring the exploration of alternative approaches. Further research is crucial to elucidate the complex relationships between LSVs and its honeybee and mite hosts. Unraveling the primary transmission routes of LSVs will equip beekeepers with critical knowledge to implement effective disease management strategies and safeguard the health of honeybee colonies. There are many variants of LSV in honeybee apiaries, and pathogenicity varies among genotypes, making it difficult to control diseases caused by LSV in apiaries. The three LSV genotypes (LSV2, LSV3, and LSV4) and the feature near-complete genome information identified in this study are important for further studies to determine the virulence of LSV in Korean apiculture and to select highly pathogenic genotypes for viral treatments such as RNA interference. The prevalence of LSV2 is closely associated with weaker colonies [23]. In the present study, SWL colonies showed a high LSV infection rate, indicating that LSV may also contribute to SWL in the ROK. However, further studies are required to understand the

interactions between LSV genotypes and other pathogens, environmental stressors, and the multifaceted, complex factors contributing to SWL and colony collapse disorder in honeybee colonies.

Analysing the RdRp nucleotide sequences of LSV2, LSV3, and LSV4, our phylogenetic trees revealed a close relationship between Korean LSV strains and those found in China. Furthermore, these Korean strains clearly differed from LSV strains obtained from the US and Belgium. These findings align with previous research by Hou et al. [37] and Kwon et al. [16], suggesting an epidemiological link between the ROK and China. Additionally, the near-completeness of our 2022 Korean LSV2, LSV3, and LSV4 strain genomes offers valuable insights into the diversity within Korean bee populations. Therefore, it is important to extend the study to the surveillance of other LSV genotypes and the impact of each genotype on beekeeping in the country.

## Supporting information

**S1 Fig. Primers designed for specific detection of LSV genotypes.** The positions of the forward and reverse primers inside the RdRp-encoding genes are marked. Sequences of each LSV genotype with NCBI accession numbers are shown.
(ZIP)

**S2 Fig. Amplification of LSV from honeybee samples using LSV2-specific primers.** The PCR products were amplified using an LSV2-specific primer pair. Lanes 1–4, 5, and 7 show the PCR products of the LSV amplification of LSV2 (218 bp); lanes 6, 8, and 9 show the bands for LSV3 (460 bp); lanes 10 and 12 show bands for LSV4 (1,000 bp); lane "-", Negative control; lane M, 100 bp DNA marker ladder (Enzynomics, Daejeon, ROK).
(TIF)

**S3 Fig. Alignment of Lake Sinai Virus 2, 3, and 4 sequences from honeybee samples.** The sequences of LSV2, LSV3, and LSV4 isolated from honeybee samples were sequenced using forward and reverse primers of LSV2. The reference sequence of each LSV genotype with its NCBI accession number is shown.
(TIF)

**S1 Table. Severe and normal winter loss information of honeybee colonies in the Republic of Korea, 2022.**
(XLSX)

**S2 Table. Primers used for detecting LSV2, LSV3, and LSV4.**
(DOCX)

**S3 Table. Primers used for determining the Lake Sinai Virus genome.**
(DOCX)

**S4 Table. Comparison of the near-complete genome feature of LSV2/Korea-2022 with reference strains in GenBank.**
(DOCX)

**S5 Table. Comparison of the near-complete genome feature of LSV3/Korea-2022 with reference strains in GenBank.**
(DOCX)

**S6 Table. Comparison of the near-complete genome feature of LSV4/Korea-2022 with reference strains in GenBank.**
(DOCX)

**S7 Table. Detection of Lake Sinai Virus in *Varroa destructor* and *Apis mellifera*.**
(DOCX)

## Acknowledgments

Our heartfelt thanks go to the beekeepers who facilitated the sampling. We extend our appreciation to all members of our laboratories for their unwavering dedication and diligent efforts.

## Author Contributions

**Conceptualization:** Thi-Thu Nguyen, Mi-Sun Yoo, Yun Sang Cho.

**Data curation:** Thi-Thu Nguyen, Mi-Sun Yoo, Yun Sang Cho.

**Formal analysis:** Thi-Thu Nguyen, A-Tai Truong.

**Methodology:** So Youn Youn, Dong-Ho Kim, Se-Ji Lee.

**Writing – original draft:** Thi-Thu Nguyen.

**Writing – review & editing:** A-Tai Truong, Soon-Seek Yoon, Yun Sang Cho.

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
