## [Decision Letter · Decision Letter 0]

24 Oct 2023

PONE-D-23-28265Prevalence and complete genome sequence of Lake Sinai virus from Apis mellifera in the Republic of KoreaPLOS ONE

Dear Dr. Cho,

Thank you for submitting your manuscript to PLOS ONE. After careful consideration, we feel that it has merit but does not fully meet PLOS ONE’s publication criteria as it currently stands. Therefore, we invite you to submit a revised version of the manuscript that addresses the points raised during the review process. While one reviewer was fully satisfied with your manuscript (which is highly unusual), the other reviewer was very critical and requests a number of improvements. I agree that your manuscript can be much stronger by considering these criticisms to improve the presentation of your interesting research.

We look forward to receiving your revised manuscript.

Kind regards,

Olav Rueppell

Academic Editor

PLOS ONE

Journal Requirements:

https://journals.plos.org/plosone/s/file?

Reviewers' comments:

Reviewer's Responses to Questions

**Comments to the Author**

1. Is the manuscript technically sound, and do the data support the conclusions?

Reviewer #1: No

Reviewer #2: Yes

2. Has the statistical analysis been performed appropriately and rigorously? 

Reviewer #1: Yes

Reviewer #2: Yes

3. Have the authors made all data underlying the findings in their manuscript fully available?

Reviewer #1: Yes

Reviewer #2: Yes

4. Is the manuscript presented in an intelligible fashion and written in standard English?

Reviewer #1: No

Reviewer #2: Yes

5. Review Comments to the Author

Reviewer #1: Summary:

The study reports a survey of Korean apiaries for strains of the LSV complex of viruses. The study does not appear to recognize the already existing set of assays or justify designing a new one, nor adequately determine the scope of that assay given the recognized high variation within the complex. The study does not sufficiently explain their sampling method and how it compares to other survey methods. Individual bees appear to have been used, even though pools of workers would generally be preferred to characterize colony-level health. Was sampling effort even among colonies and apiaries? Were colonies chosen in a randomized fashion from around the country? Some context as to the commercial/landscape status of the apiaries that were surveyed would be helpful, as these factors are important in colony health and disease transmission.

The reporting of the work needs improvement in several aspects. First, the authors are inconsistent and inaccurate in describing their survey target, namely LSV2 based on a single accession and without regard for other primers already in the literature. The survey prevalence of LSV3 and LSV4 types do not appear to be based on targeted primer design but on accidental amplification of those types with the LSV2 primers. Second, the prevalence data would be better presented in a geographic context (ie a map of pie charts of detection rates) and the sequence similarity data is excessive with little apparent thought given to the utility of what is being presented and why.

Specific comments:

1. Title must be modified to reflect the actual scope of LSV survey (eg LSV2), such as Prevalence and complete genome sequence of selected strains of Lake Sinai virus from Apis mellifera in the Republic of Korea. Also, how can the genomes be complete if they were generated by amplification and overlap? There is no way of knowing what sequence might be beyond the primers.

2. Line 55: The fact that nomenclature within LSV is ad hoc and does not necessarily reflect evolutionary relationships should be noted. To what extent lineages are evolutionary units comparable to species, and whether the polymerase or the whole genome should be used for phylogenetics, is not really resolved. ViralZone uses only LSV1 and LSV2, for example, whereas NCBI uses an inconsistent nomenclature (submitted by users) of numerous numbered and lettered strains. The strains were named essentially sequentially, rather than based on any comprehensive analysis of global sequence variation. LSV1 and LSV2 appear to be deep divisions within the overall variation, with other named types falling nearer one or the other of these two clades.

3. Line 53: Most authors identify four ORFs, see Bigot et al. While ORF4 has unusual properties and is not in all LSV relatives and may even be absent from some LSV, it should still be mentioned (see Shi et al., Bigot et al., Cornman, Karlin; see also https://www.uniprot.org/uniprotkb/?query=xref:embl-KX883223)

4. Lines 43-45: The first part of this sentence should be deleted as the association with colony health has already been noted. Instead, the point should just be that LSVs were first discovered in US surveys and subsequently demonstrated to be widespread in US.

5. Line 52: not accurate, many accessions are >6kb.

6. Line 63: There is no direct evidence that “LSV may cause behavioral changes in honeybees”, certainly not in the references cited for this statement.

7. Lines 66-73: this paragraph is redundant and unnecessary.

8. Line 82: “sample” does not appear to be defined. Individual worker bees? Of a particular age/caste/activity?

9. Line 97: what was the rationale for this primer design? Seems to be targeting LSV2 specifically. Why not based on an alignment of multiple LSV accessions to determine specificity? Most studies have had to use primers targeting a subset of LSV lineages due to the variability of the complex (but see Iwanowicz et al. 2020, https://doi.org/10.7717/peerj.9424).

10. Line 118: Ct < 35 is probably reasonable, but a bit arbitrary. Do the Ct values for negative and positive controls support this choice?

11. Line 127: this makes no sense as an a priori strategy. Presumably the initial amplicons were found to closely match the listed accessions rather than “LSV2”, and thus this strategy was undertaken? Please explain.

12. Line 136: searched against what database? NCBI nt database?

13. Line 139: “gaps were edited and calculated” means what? Manually aligned after initial clustal alignment?

14. Lines 259-268: At least two studies (Iwanowicz et al. 2020 and McMenamin et al. 2021) have detected LSV in other bee species as well, there may be additional reports I’m not aware of.

15. Line 149: LSV2 was targeted by primer design, but cross-amplification with other lineages was evident in the recovery of LSV3 and LSV4-like sequences. So language that reflects the limited and uncertain scope of the LSV assay needs to be used. For example, one could say “the LSV2 PCR assay was positive” rather than “LSV was present”.

16. Please show the results in their spatial context, ie a map with pie charts for each location rather than as bar plots.

17. Line 161: Earlier on line 138, the analyzed region was stated to be 460nt.

18. Line 176: what primers were used to determine LSV3 and LSV4 rates? Surely not the LSV2 primer set described in the methods, that produced the unexpected large amplicon? Primers should be designed for the purpose or else the results are potentially biased.

19. Line 82: it would be helpful to characterize the colonies in terms of use. Are these used for pollination services or honey? Are they transported during the year? Hobbyists, small commercial apiaries, large commercial apiaries? Agricultural environment?

20. I do not see the value of tables of percent identity among various accessions. Move to supplemental materials.

21. Line 252: all these cited studies used different sampling techniques, different assays, and different criteria for positives. There is no basis for directly comparing prevalence rates of the LSV complex by country, or at least the authors do not justify doing so. Nor are country borders particularly relevant to virus dispersal. Factors like economic trade zones, continents or biogeographic regions, ranges of mellifera subspecies, apicultural practice etc are probably more relevant.

Reviewer #2: This is well written manuscript, interesting for publication. The Lake Sinai virus is emerging pathogen and valuable informations are presented in this manuscript with prevalence of diffrent genotypes, related to the severe or normal honeybee winter colonies losses. The manuscript is including also complete genomes of LSV2, LSV3 and LSV4 from Republic of Korea and comparison to the published sequences.

I have no remarks. The manuscript is suitable for publication in the present form.

6. PLOS authors have the option to publish the peer review history of their article (what does this mean?). If published, this will include your full peer review and any attached files.

Reviewer #1: No

Reviewer #2: No

---

## [Author Response · Author response to Decision Letter 0]

19 Dec 2023

Responses to Reviewers’ Comments

We appreciate the reviewers for their invaluable comments. As explained below, we have revised our original manuscript and materials in response to all of the reviewers’ comments. What follows are our point-by-point responses to the comments.

Reviewer #1: Summary:

The study reports a survey of Korean apiaries for strains of the LSV complex of viruses. The study does not appear to recognize the already existing set of assays or justify designing a new one, nor adequately determine the scope of that assay given the recognized high variation within the complex. The study does not sufficiently explain their sampling method and how it compares to other survey methods. Individual bees appear to have been used, even though pools of workers would generally be preferred to characterize colony-level health. Was sampling effort even among colonies and apiaries? Were colonies chosen in a randomized fashion from around the country? Some context as to the commercial/landscape status of the apiaries that were surveyed would be helpful, as these factors are important in colony health and disease transmission.

The reporting of the work needs improvement in several aspects. First, the authors are inconsistent and inaccurate in describing their survey target, namely LSV2 based on a single accession and without regard for other primers already in the literature. The survey prevalence of LSV3 and LSV4 types do not appear to be based on targeted primer design but on accidental amplification of those types with the LSV2 primers. Second, the prevalence data would be better presented in a geographic context (ie a map of pie charts of detection rates) and the sequence similarity data is excessive with little apparent thought given to the utility of what is being presented and why.

Specific comments:

1. Title must be modified to reflect the actual scope of LSV survey (eg LSV2), such as Prevalence and complete genome sequence of selected strains of Lake Sinai virus from Apis mellifera in the Republic of Korea. Also, how can the genomes be complete if they were generated by amplification and overlap? There is no way of knowing what sequence might be beyond the primers.

Response:

Thank you for your insightful comments on our article's title. We highly appreciate your feedback and concur that the title needs to be revised to accurately reflect the scope of the LSV survey on Korean honeybees.

According to previous studies the LSV2 was the most virulent genotype. To identify the reason of serious colony loss in South Korea we initially select LSV2 as the target for detection and identification. However, after analyzing sequencing result we revealed that three genotypes, LSV2, LSV3 and LSV4 existed in Korean apiaries. Therefore, we finally designed the specific primer pair for each LSV genotype (S2 Table), and the surveillance data of each LSV genotype was acquired using genotype specific primer pair. To identify the complete genome sequence of LSV the different primer pairs were designed on the conserved region of reference sequences of each genotype on NCBI, different fragments had overlap parts that were used for assembly of the whole genome. 

The objective of our study is to detect LSV variants in honeybees collected from the Republic of Korea (ROK) and assess the prevalence of this virus in samples with information on winter colony loss compared to normal colonies. In addition, our research aims to comprehensively analyze the genomic features of LSV isolates from Korean honeybees in comparison to strains previously identified and isolated in different countries. These insights will provide valuable guidance for future studies aimed at enhancing the health of honeybee apiaries in the ROK.

I propose the revised title: "Prevalence and Genomic Features of Lake Sinai Virus Isolated from Apis mellifera in the Republic of Korea”.

2. Line 55: The fact that nomenclature within LSV is ad hoc and does not necessarily reflect evolutionary relationships should be noted. To what extent lineages are evolutionary units comparable to species, and whether the polymerase or the whole genome should be used for phylogenetics, is not really resolved. ViralZone uses only LSV1 and LSV2, for example, whereas NCBI uses an inconsistent nomenclature (submitted by users) of numerous numbered and lettered strains. The strains were named essentially sequentially, rather than based on any comprehensive analysis of global sequence variation. LSV1 and LSV2 appear to be deep divisions within the overall variation, with other named types falling nearer one or the other of these two clades.

Response:

We agree that the nomenclature used in the LSV study is particular and doesn't always accurately reflect evolutionary relationships. Based on phylogenetic analyses of RdRp, two common clades have been identified, termed LSV1 and LSV2. We acknowledge the inconsistency in nomenclature used by various databases, such as ViralZone and NCBI, as well as the lack of comprehensive sequence variant analysis guiding nomenclature. Additionally, other strains labeled as LSV have been documented.

The sentence will be revised as follows: Continuous molecular surveys are underway to assess the relative conservation of various LSV strains due to the significant diversity observed among them. Presently, species-specific RdRp phylogenetic analyses have led to the identification of two phylogenetic clusters known as LSV1 and LSV2 [7]. Furthermore, there exist strains labeled as sister strains, potentially representing recombinants of LSV1 and LSV2, with designations on the National Center for Biotechnology (NCBI) such as LSV3–LSV8, LSV NE, LSV SA1, LSV SA2, LSV TO. Discrimination among these strains relies heavily on molecular surveys, including PCR-based methods, RT-PCR, real-time RT-PCR, and metagenomics. It is noteworthy that the primers utilized for variant detection have, thus far, not exhibited sensitivity to all known gene types [3,4,7,9,22]. (Lines 57-66)

3. Line 53: Most authors identify four ORFs, see Bigot et al. While ORF4 has unusual properties and is not in all LSV relatives and may even be absent from some LSV, it should still be mentioned (see Shi et al., Bigot et al., Cornman, Karlin; see also https://www.uniprot.org/uniprotkb/?query=xref:embl-KX883223)

Response:

We appreciate the references you've provided, including Bigot et al. and Shi et al., which discuss the unique properties of ORF4 and its presence in some LSV relatives. While our study primarily focused on the key ORFs in LSV, we acknowledge that ORF4 is indeed an intriguing and less common feature in LSV relatives. Its variable presence among different LSV strains is a subject of interest and further investigation. We understand the importance of mentioning ORF4, and we apologize for not including it in our initial discussion.

The sentence will be revised as follows: The genome characteristics of LSV encompass three main genes: 1) the first non-structural protein (NS1), 2) RNA-dependent RNA polymerase (RdRp), and 3) Capsid protein (CP) [4,7,19,20]. Additionally, the genome contains a region for the second non-structural protein (NS2) with distinctive features that may be absent in LSV clades. The NS1 and RdRp regions of LSV are implicated in the evolutionary process [5]. The capsid region plays a crucial role in the virus lifecycle, as it recognizes the host and assembles the virus capsid [19]. Meanwhile, the NS2 region (ORF4) can facilitate arthropod infection [21]. (Lines 50-57)

4. Lines 43-45: The first part of this sentence should be deleted as the association with colony health has already been noted. Instead, the point should just be that LSVs were first discovered in US surveys and subsequently demonstrated to be widespread in US.

Response: Modified as suggested. 

LSVs were first discovered in the United States (US) surveys [2,7] and subsequently demonstrated to be widespread in the US [4,8,9]. (Lines 42-44)

5. Line 52: not accurate, many accessions are >6kb.

Response: Upon re-evaluating our data, we have identified instance where many segments are indeed larger than 6 kb. We will correct this error in our revised draft to ensure the accuracy of our findings. Additionally, we will focus our discussion only on the genes present in the LSV genome.

6. Line 63: There is no direct evidence that “LSV may cause behavioral changes in honeybees”, certainly not in the references cited for this statement.

Response: Modified as suggested.

The sentence was revised as follows: Due to the lack of symptomatic features of LSV in honeybees, molecular biological diagnosis becomes imperative for detecting the infection levels of this virus within honeybee apiaries. (Lines 70-72).

7. Lines 66-73: this paragraph is redundant and unnecessary.

Response: The paragraph was eliminated from the manuscript. 

8. Line 82: “sample” does not appear to be defined. Individual worker bees? Of a particular age/caste/activity?

Response: 

The worker bees (n=10~30) were collected from each of 266 colonies in 137 apiaries in the ROK. The information is added in the manuscript in Line 90. To collect the sample, the hive of each colony was opened, and then the comb was taken out after carefully observing the comb without queen bee. Ten worker bees were randomly captured using a 50mL conical tube from the comb. In the colonies that all bees are dead, 10 worker bees were collected at the bottom of the hive. 

9. Line 97: what was the rationale for this primer design? Seems to be targeting LSV2 specifically. Why not based on an alignment of multiple LSV accessions to determine specificity? Most studies have had to use primers targeting a subset of LSV lineages due to the variability of the complex (but see Iwanowicz et al. 2020, https://doi.org/10.7717/peerj.9424).

Response:

The primer pair amplification and analysis of LSV RdRp region was designed from multi sequence alignment. To do that, the LSV2 sequences (NCBI No.: HQ888865.2) was used for nucleotide blast on NCBI, and then the sequences with high identity were selected for alignment to design the primer pair. 

10. Line 118: Ct < 35 is probably reasonable, but a bit arbitrary. Do the Ct values for negative and positive controls support this choice?

Response:

The positive samples utilized in this study were amplified from Korean honeybees, subsequently cloned into pGem–T vector and transferred to E. coli DH5α cells. The Ct value of positive detection was decided based on the limit detection using recombinant plasmid of each LSV genotype and the result of negative control without DNA template. The lowest DNA copy number of LSV2, LSV3, and LSV4 that the qPCR can detect were 5.10^1, 8.10^1, 4.10^1 copy with Ct value 38.32±1.99, 36.38±1.2, and 38.23±1.15, respectively. The negative control showed the Ct value after 39.08±1.59 cycle with LSV3, and could not detected with LSV2 and LSV4. Hence, we selected the Ct value ≤ 35 to detect the presence of LSV in the samples in this study. This threshold allows us to identify positive samples reliably while minimizing the risk of false positives.

11. Line 127: this makes no sense as an a priori strategy. Presumably the initial amplicons were found to closely match the listed accessions rather than “LSV2”, and thus this strategy was undertaken? Please explain.

Response:

After identifying the LSV2, LSV3, and LSV4 based on RdRp gene using the common primer pair LSV-For: 5′- GCT TGT CGT GGA TTC TGG TC -3′ and LSV-Rev:5′-CTC AGC ACG AAA TCG CTC AA-3′, other the specific primer pairs for segment amplification of whole genome of each LSV genotype was designed (S3 Table). The various fragments amplified from each LSV genotype were assembled by comparing to the reference genome sequence of each genotype.

12. Line 136: searched against what database? NCBI nt database?

Response: 

The sentence will be revised as follows: The LSV sequences were analyzed using the NCBI nucleotide Basic Local Alignment Search Tool to identify related sequences. (Lines 153-154).

13. Line 139: “gaps were edited and calculated” means what? Manually aligned after initial clustal alignment?

Response:

After alignment using the Clustal program the aligned result showed some sequence acquired from NCBI has longer size than the generated sequences in this study. Therefore, the program Bioedit was used to trim the redundant overhang part of the reference sequence before using for the phylogenetic creating by the Mega X program.

14. Lines 259-268: At least two studies (Iwanowicz et al. 2020 and McMenamin et al. 2021) have detected LSV in other bee species as well, there may be additional reports I’m not aware of.

Response:

Thank you for highlighting the studies conducted by Iwanowicz et al. (2020) and McMenamin et al. (2021), which detected LSV in other bee species. We appreciate your valuable input, and we will incorporate your suggestions into the introduction section.

The sentence will be revised as follows: LSV is not only detected in honey bees but also has been identified in certain ant species [20], on pollen loads and Varroa destructor [4], and wild bumblebee hosts (namely Bombus pascuorum, Bombus lapidaries, Bombus pratorum, Bombus atratus [24,25], sweat bee (Halictus ligatus) [9], solitary bees (Andrena vaga, Osmia bicornis, Osmia cornuta) [26], mining bee (Andrena spp.) [3,27], and Vespids (Hornet) [28]. (Lines 73-77)

15. Line 149: LSV2 was targeted by primer design, but cross-amplification with other lineages was evident in the recovery of LSV3 and LSV4-like sequences. So, language that reflects the limited and uncertain scope of the LSV assay needs to be used. For example, one could say “the LSV2 PCR assay was positive” rather than “LSV was present”.

Response: 

The sentence will be revised as follows: The LSV2 PCR assay was positive in honeybees, with a high infection rate of 60.9 % (162/266) in the ROK. A comparison of the presence of LSV2 in different provinces of the ROK revealed a statistically significant variation in LSV2 infection rates (p = 0.011). In particular, a high prevalence of LSV2 was observed in Gyeongsangbuk-do, Gyeongsangnam-do, and Jeollanam-do, with infection rates of 32.5, 21.6, and 17.3%, respectively. (Lines 149-153)

16. Please show the results in their spatial context, ie a map with pie charts for each location rather than as bar plots.

Response: Modified as suggested.

Fig 2. Infection rate of the Lake Sinai Virus 2 (LSV2) isolated from honeybee samples between January and August 2022, according to sampling provinces and cities in the Republic of Korea.

17. Line 161: Earlier on line 138, the analyzed region was stated to be 460nt.

Response: Modified as suggested. the length of RdRp region (460 nt) is now using in the sentence 

18. Line 176: what primers were used to determine LSV3 and LSV4 rates? Surely not the LSV2 primer set described in the methods, that produced the unexpected large amplicon? Primers should be designed for the purpose or else the results are potentially biased.

Response:

The LSV2 primer set described in the methods was not specifically designed to target LSV3 and LSV4, and it indeed produced an unexpectedly large amplicon. To address this concern and ensure the accuracy of our results, we have undertaken the redesign of two new primer pairs (detailed in Supplementary S1-Fig and S2 Table), that was mentioned in Line 113 of the method section. These genotype specific primers were used for wide surveillance of the LSV in apiaries. 

19. Line 82: it would be helpful to characterize the colonies in terms of use. Are these used for pollination services or honey? Are they transported during the year? Hobbyists, small commercial apiaries, large commercial apiaries? Agricultural environment?

Response: We agree that providing more information about the colonies would be beneficial for a comprehensive understanding of the study. The colonies in our study were sourced from various types of apiaries across different provinces in ROK. Beekeepers are primarily situated in mountainous forested regions, capitalizing on abundant natural resources. These honeybee colonies in apiaries are chiefly dedicated to honey harvesting, typically remaining fixed at a specific location. Colony number in each apiary ranged from 50 to 100.

The honeybee samples in this study were collected based on the observed decline in the vitality of adult honeybees and an unusual decrease in the population. The symptoms within honeybee colonies corresponded to the phenomenon of severe winter loss or Colony Collapse Disorder (CCD). The number of honeybee samples collected depended on the number of lost honeybee colonies in each apiary. (Lines 91-97)

20. I do not see the value of tables of percent identity among various accessions. Move to supplemental materials.

Response: Thank you for your suggestion. The Table 1–3 were moved to Supplementary materials as S4–S6 Tables. A phylogenetic tree comparing the features of genomes isolated from Korean honeybees was newly added in the manuscript as Fig 6.

Fig 6. Neighbor-joining phylogenetic tree of Lake Sinai Virus based on the feature genome. The optimal tree is shown. The evolutionary distances were computed using the Kimura 2-parameter method. A, LSV2 genome feature (5,759 nt); B, LSV3 genome feature (6,040 nt); C, LSV4 genome feature (5,985 nt). (References strains in NCBI were shown in S4–S6 Tables).

The phylogenetic variation is consistent with nucleotide similarity among the isolates. In the first main branch, the genome sequences of LSV2, LSV3 and LSV4 isolate from Korean honeybees were similarity with group isolate to China. As shown in the tree, the LSV isolates from ROK had a closer genetic relationship with a species of LSV isolated from China (Fig 6). (Lines 221-224)

21. Line 252: all these cited studies used different sampling techniques, different assays, and different criteria for positives. There is no basis for directly comparing prevalence rates of the LSV complex by country, or at least the authors do not justify doing so. Nor are country borders particularly relevant to virus dispersal. Factors like economic trade zones, continents or biogeographic regions, ranges of mellifera subspecies, apicultural practice etc are probably more relevant.

Response: Modified as suggested.

The LSV was first detected and described in honeybee colonies in the US [7]. Subsequently, its occurrence has been reported about the prevalence of LSV in honeybee populations in many countries, including Germany, Australia, Slovenia, the Czech Republic, Africa, Iran, and Japan [4,11,12,15,18,35,36]. (Lines 258-261)

---

## [Decision Letter · Decision Letter 1]

5 Jan 2024

PONE-D-23-28265R1Prevalence and Genome Features of Lake Sinai Virus Isolated from Apis mellifera in the Republic of KoreaPLOS ONE

Dear Dr. Cho,

Thank you for submitting your manuscript to PLOS ONE. After careful consideration, we feel that it has merit but does not fully meet PLOS ONE’s publication criteria as it currently stands. Therefore, we invite you to submit a revised version of the manuscript that addresses the points raised during the review process.

 Please carefully consider the thorough re-evaluation of the reviewer who was initially very critical. While progress has been noted, several serious issues remain to be resolved.

We look forward to receiving your revised manuscript.

Kind regards,

Olav Rueppell

Academic Editor

PLOS ONE

Journal Requirements:

Reviewers' comments:

Reviewer's Responses to Questions

**Comments to the Author**

1. If the authors have adequately addressed your comments raised in a previous round of review and you feel that this manuscript is now acceptable for publication, you may indicate that here to bypass the “Comments to the Author” section, enter your conflict of interest statement in the “Confidential to Editor” section, and submit your "Accept" recommendation.

Reviewer #1: (No Response)

2. Is the manuscript technically sound, and do the data support the conclusions?

Reviewer #1: Partly

3. Has the statistical analysis been performed appropriately and rigorously? 

Reviewer #1: I Don't Know

4. Have the authors made all data underlying the findings in their manuscript fully available?

Reviewer #1: Yes

5. Is the manuscript presented in an intelligible fashion and written in standard English?

Reviewer #1: Yes

6. Review Comments to the Author

Reviewer #1: Summary: I appreciate the authors’ efforts to clarify their methodology and add context to their Introduction and Discussion. I still feel that the potential for primer bias is not fully acknowledged, despite the clear potential evident in Supplemental Figure S1, for example. The authors may have felt that the LSV2 accession used for primer design well represented previous results in the ROK, and they could also cite Hou et al. 2023 https://doi.org/10.1111/mec.16987, which shows continental genetic clustering within all the major LSV clades, as justification for their approach. Nonetheless, the paper makes many statements that gloss over ascertainment issues, such as in the abstract: “This study aimed to assess the prevalence of ... the LSVs ...from the ROK”). These claims overlook the fact that LSV1 is not assayed and assays for LSV2/LSV3/LSV4 have known primer mismatches with LSV2, LSV3, or LSV4 clade members. See also lines 81-86, line 109, and line 251 as examples where LSV is used instead of the specific LSV genotypes for which an assay was developed.

Some additional comments are provided below.

The authors continue to state that they have generated “complete genome sequences of LSV” using primers that do not start at position 1 of any reference and do not end at the final position of any reference. It is a minor point of no import for any conclusions, but these overlapped sequences cannot be called “complete” genomes, even if 99% of the bases are captured by the method. Line 221, for example, would need to be changed to “near-complete genome” or “partial genome”. Similarly, in the abstract the phrase “the first genome feature of the Korean “ should be changed to “the first near-complete genome sequences of Korean”. Nor is it certain whether the overlapped amplicons all come from a single virus strain, ie the assemblies could be chimeric. It has been previously documented that single individuals can harbor diverse strains (e.g. Ravoet et al; Iwanowicz et al.).

Lines 53-54: Not correct as written, change to “with distinctive features that may be absent in some LSV clades”? Also, does “the evolutionary process” mean “replication” (line 54)?

Line 65-66: “genotypes” rather than “gene types”

The authors continue to state “LSV may cause behavioral changes in honeybees, leading to worker bees abandoning their hives” (lines 77-78). The cited references only show that LSV was found to be more prevalent in weak/collapsed hives than healthy. Nothing can be said about “behavioral changes in honeybees” caused by LSV. The fact that there may be a behavorial component to the colony collapse phenomenon does not mean LSV causes any behavior.

Lines 142-143: I think “The alignment was generated and trimmed to the RdRp gene region in Bioedit” would be clearer.

Lines 152-153: The BLASTN software is different from the database it was used to search. The default for BLASTN for the NCBI web service is the nt database, but any database can be used in principle. If nt was used, please explicitly state so for completeness/repeatability.

Line 162: As the ANOVA is performed on proportions of positives within colonies, have the authors verified that the data are reasonably distributed? If not, a Kruskal-Wallis nonparametric test could be used instead.

Line 166: “detection rate” instead of “infection rate”

Line 168-170: if the overall detection rate of LSV2 was greater than 60%, how can the values on line 170 be considered high?

Lines 176-178: the %id is lower for the LSV2 accession than the LSV3 and LSV4 accessions, in reverse order to what the text states.

Line 182: this could just be due to primer bias – European like lineages might be present in ROK apiaries but not amplified.

Line 274: What was the nature of the varroa samples? Were they collected from sticky boards, sugar roll, directly from parasitized adults or pupae? Were individual mites analyzed or pools? How were the twelve colonies selected for this assay?

Also, what is your conclusion regarding the fact that your LSV4 assay was never positive in mites, although numerous positives in honeybees were obtained? Are patterns of discordance between the two species in S7 Table a matter of chance sampling or do you believe some species-level difference in LSV host interaction is occurring? Some discussion would be appreciated.

Line 288-290: should note this is consistent with results of Hou et al. 2023, but also potentially an artifact of primer bias.

7. PLOS authors have the option to publish the peer review history of their article (what does this mean?). If published, this will include your full peer review and any attached files.

Reviewer #1: No

---

## [Author Response · Author response to Decision Letter 1]

8 Feb 2024

I attach the revised manuscript with track change and raw image file as one pdf file (S1-raw-images) according to the points by editor.

Sincerely yours,

Yun Sang Cho

---

## [Editor Report · Decision Letter 2]

13 Feb 2024

Prevalence and Genome Features of Lake Sinai Virus Isolated from Apis mellifera in the Republic of Korea

PONE-D-23-28265R2

Dear Dr. Cho,

We’re pleased to inform you that your manuscript has been judged scientifically suitable for publication and will be formally accepted for publication once it meets all outstanding technical requirements.

Kind regards,

Olav Rueppell

Academic Editor

PLOS ONE
---

## [Editor Report · Acceptance letter]

4 Mar 2024

PONE-D-23-28265R2 

PLOS ONE

Dear Dr. Cho, 

I'm pleased to inform you that your manuscript has been deemed suitable for publication in PLOS ONE. Congratulations! Your manuscript is now being handed over to our production team.

Kind regards, 

on behalf of

Dr. Olav Rueppell 

Academic Editor

PLOS ONE